# Establishment and Characterization of Novel Human Intestinal In Vitro Models for Absorption and First-Pass Metabolism Studies

**DOI:** 10.3390/ijms23179861

**Published:** 2022-08-30

**Authors:** Randy Przybylla, Christina Susanne Mullins, Mathias Krohn, Stefan Oswald, Michael Linnebacher

**Affiliations:** 1Molecular Oncology and Immunotherapy, Clinic of General Surgery, 18057 Rostock, Germany; 2Institute of Pharmacology and Toxicology, Rostock University Medical Center, 18057 Rostock, Germany

**Keywords:** intestinal epithelial models, barrier funtion, first-pass metabolism, drug absorption and metabolism

## Abstract

Commonly used intestinal in vitro models are limited in their potential to predict oral drug absorption. They either lack the capability to form a tight cellular monolayer mimicking the intestinal epithelial barrier or the expression of cytochrome P450 3A4 (CYP3A4). The aim of this study was to establish a platform of colorectal cancer patient-derived cell lines for evaluation of human intestinal drug absorption and metabolism. We characterized ten 2D cell lines out of our collection with confluent outgrowth and long-lasting barrier forming potential as well as suitability for high throughput applications with special emphasis on expression and inducibility of CYP3A4. By assessment of the transepithelial electrical resistance (TEER) the cells barrier function capacity can be quantified. Very high TEER levels were detected for HROC60. A high basal CYP3A4 expression and function was found for HROC32. Eight cell lines showed higher CYP3A4 induction by stimulation via the vitamin D receptor compared to Caco-2 cells (5.1- to 16.8-fold change). Stimulation of the pregnane X receptor led to higher CYP3A4 induction in two cell lines. In sum, we identified the two cell lines HROC183 T0 M2 and HROC217 T1 M2 as useful tools for in vitro drug absorption studies. Due to their high TEER values and inducibility by drug receptor ligands, they may be superior to Caco-2 cells to analyze oral drug absorption and intestinal drug–drug interactions. Significance statement: Selecting appropriate candidates is important in preclinical drug development. Therefore, cell models to predict absorption from the human intestine are of the utmost importance. This study revealed that the human cell lines HROC183 T0 M2 and HROC217 T1 M2 may be better suited models and possess higher predictive power of pregnane X receptor- and vitamin D-mediated drug metabolism than Caco-2 cells. Consequently, they represent useful tools for predicting intestinal absorption and simultaneously enable assessment of membrane permeability and first-pass metabolism.

## 1. Introduction

In the preclinical phases of drug discovery, cell-based in vitro models of the human intestine are necessary tools for ADME (absorption, distribution, metabolism, excretion) property assessment of new molecular entities. Poor oral absorption and disadvantageous metabolic properties are frequent reasons for failure of novel drug candidates [1]. For decades, the cell line Caco-2 has been widely used as an in vitro model of the human intestinal barrier to study ADME and toxicity. Originally established from a human colon carcinoma, this cell line spontaneously differentiates in culture forming an epithelial cell monolayer consisting of absorptive enterocyte-like cells [2,3]. When reaching confluence, Caco-2 cells form microvilli on their apical side and are coupled by tight junctions, the so-called brush border [4]. A generally accepted surrogate for the assessment of cellular barrier integrity and tight junction dynamics is the transepithelial electrical resistance (TEER) measurement [5]. The TEER value is a reflection of the ionic conductance of the paracellular pathway in the epithelial monolayer [6]. For compounds, both passively absorbed and actively transported, the drug permeability across Caco-2 monolayers correlates well with the percentage of molecules absorbed in the human intestine [7,8]. Despite extensive application in drug discovery, several limitations of the Caco-2 cell monolayer model prevail. Heterogeneity causes extensive variation; several clones with significant variations in gene expression and transport activities have been described [3,9,10]. Compared to the permeability of the human small intestine, Caco-2 cell layers are less pervious for hydrophilic drugs absorbed via the paracellular route [11,12]. Even more importantly for drug development, cytochrome P450 (CYP) 3A4 is barely expressed in Caco-2 cells, restricting its utility of analyses involving CYP3A4-mediated drug metabolism and CYP3A4 drug induction potential [13,14]. Since CYP3A4 plays a crucial role in the enterocytic biotransformation of endobiotics and xenobiotics [15], it is the single most important factor in oral drug bioavailability [16,17].

Shortcomings of other established colorectal cancer (CRC) cell lines such as HT-29, LS180, and T84 are deficient tight junction formation and improper CYP3A4 expression or inducibility [18,19]. Primary fetal human small intestinal epithelial cells only have low basal CYP3A4 activity and additionally lack induction potential [19]. Thus, human-induced pluripotent stem cell-derived enterocyte-like cells were reported as a more appropriate model for drug-mediated CYP3A4 induction tests [20]. However, this elaborate technique is comparably complex, time intensive, and expensive.

The aim of the present study was to identify and characterize 2D in vitro models better suited for intestinal drug permeability and first-pass metabolism studies with superior physiological properties than the Caco-2 cell line. Therefore, the large number of low-passaged human CRC cell lines of the HROC collection was taken advantage of [21].

## 2. Results and Discussion

The most commonly used models in drug discovery research are still 2D cell cultures such as Caco-2 [22]. In this study, we present a panel of 2D-growing, low-passaged human intestinal cell models meeting basic requirements essential for intestinal models including the ability to grow as long-lasting and barrier-forming cell layers [23] and applicable for the prediction of oral drug absorption and bioavailability. The starting point was the HROC collection, currently encompassing > 100 patient-derived CRC cell lines, which are immortal and suitable for long-term passaging [21]. We directly compared characteristics of HROC cells to Caco-2 cells and applicability of standard in vitro assays frequently used in preclinical candidate drug testing. First, we analyzed the capacity to develop a confluent and over time stable cell monolayer as well as cell migration potential in wound healing assay (Figure 1). A confluent monolayer (95–100%) was achieved in 45 of the 49 HROC cell lines screened. In four cell lines, gaps within the layer evolved over time and in another line strong dome formation was observed in long-term cultures. Further, two cell lines peeled off as a layer within one to two days after reaching confluence. Another three lines were found not to be suitable for wound healing assays and a loss of barrier integrity was observed in one line. Finally, one cell line with a very slow proliferation rate (doubling time of > 72 h) was excluded due to this unhandy property. For the directly patient-derived and patient-derived xenograft-derived cell line pair HROC383 and HROC383 T0 M2, only the better performing partner HROC383 was included into the final panel for investigation. In total, 32 of the 49 HROC cell lines met the initial requirements and were further tested (for an overview see Appendix A). Monolayer integrity was subsequently assessed by TEER measurement over a course of 21 days. Of the 32 HROC lines tested, 22 did not reach maximum TEER levels of ≥ 300 Ω·cm^2^ and were excluded from further analysis. The time-dependent TEER development of the remaining ten HROC lines is given in Figure 2. These lines developed TEER levels ≥ 300 Ω·cm^2^ and TEER values increased over time. Considerable differences in TEER levels became evident. HROC383 showed the lowest (812 Ω·cm^2^ at day 15), HROC60 an exceptionally high maximum TEER level (6066 Ω·cm^2^ at day 17). Such high TEER values differ considerably from TEER figures reported for other CRC cell lines [5]. Of note, the HROC60 cell line is even capable of maintaining its monolayer with further TEER elevation up to > 11,500 Ω·cm^2^ after five weeks in transwell inserts. Thus, HROC60 most probably is a model superior to Caco-2 for high throughput investigations of drug transport or changes in TEER levels in drug toxicity studies over long time periods [24]. ‘Gut-on-a-chip’ investigations might also benefit from this long-term stability of functional tight junctions [25]. TEER levels of Caco-2 cells measured as reference ranged from 419 Ω·cm^2^ (at day 7, which was the first day a level > 300 Ω·cm^2^ was observed) to 1259 Ω·cm^2^ (maximum, day 16), which is fairly consistent with published data [26,27]. It should be also noted that the TEER values of some HROC cells obtained in this study are several fold-higher than TEER figures reported for human small intestine [28]. Collectively, seven of the ten tested HROC lines evinced TEER values similar or superior to Caco-2 (Table 1). Fastest establishment of TEER values > 300 Ω·cm^2^ was observed in HROC60 and HROC80 T1 M1 (first day of TEER > 300 Ω·cm^2^ was day four and day five, respectively). This is two to three days faster than the reference cells Caco-2. Barrier formation occurred latest in HROC159 T2 M4 (TEER > 300 Ω·cm^2^ at day 14) and HROC43 (TEER > 300 Ω·cm^2^ at day 12). As surrogate measure for intestinal drug permeability via paracellular transportation, the capacity of cell layers to retain FD-4 is considered to be more relevant than TEER values. Thus, we next assessed the time dependent FD-4 permeability (Figure 3). In all ten HROC lines tested, FD-4 permeability decreased over time and the lines thus fulfilled the requirement of low paracellular permeability. A very rapid decrease in permeability was observed in HROC60 and HROC80 T1 M1 cells. As expected, FD-4 permeability negatively correlated with the TEER levels (see also Table 1). Both parameters are essential for intestinal models and confirm functional tight junction formation, which is a prerequisite of intestinal barrier function [29].

Cell proliferation was determined next within passage numbers 25 to 45 (Appendix A). Proliferation assays revealed doubling times < 70 h for all HROC cell lines. In two lines, doubling time was lower than in Caco-2 cells (43.9 h for HROC183 T0 M2 and 44.5 h for HROC126 vs. 52.1 h for Caco-2). Doubling rates approximated to values of Caco-2 cells in two HROC lines (50.9 h for HROC239 T0 M1, 52.4 h for HROC80 T1 M1). Higher cell doubling times were determined for HROC60, HROC217 T1 M2, and HROC43 (ranging from 64.5 to 68.4 h).

Cell migration rate was analyzed by wound healing assays. Wound gaps were closed by all cell lines investigated (Appendix A and Figure 4). However, the duration to gap closing constituting the migration speed, varied greatly. Fastest healing, comparable to Caco-2 cells (7.7 µm/h), was observed for HROC60 (6.2 µm/h). Intermediate healing speed was observed for HROC159 T2 M4, HROC383, and HROC126 (4.7 µm/h to 4.4 µm/h), whereas the cell layer wound closed very slowly (0.7 µm/h) in the cell model HROC239 T0 M1.

Data on cancer driver mutations have been published previously [21] and are recapitulated for the final panel of HROC models as Appendix A.

In order to analyze the capacity of intestinal first-pass metabolism, we than tested the basal CYP3A4 activity of the ten HROC lines. Basal activity of HROC32, HROC43, and HROC159 T2 M4 were 6.5-, 2.8-, and 1.8-fold higher than Caco-2 basal activity (Figure 5A). CYP3A4 activity levels comparable to Caco-2 cells were observed for HROC217 T1 M2, HROC383 and HROC239 T0 M1. Lower CYP3A4 activity levels were observed for HROC126, HROC80 T1 M1, HROC183 T0 M2, and HROC60 (Figure 5A and Table 1). Tumoral CYP3A4 expression is proposed as a marker for treatment response [30]. In CRC, CYP3A4 overexpression plays a role in the inactivation of irinotecan and other xenobiotics. It is responsible for a poor response to standard chemotherapy [30,31]. There is still a necessity to identify CRC cell lines with a high basal CYP3A4 activity which mirror a poor response to drug treatment. Due to the high basal CYP3A4 activity and poor CYP3A4 inducibility, HROC32 provides a utile tool for the development of alternative drug regimens, especially for tumors with high CYP3A4 expression. Moreover, these HROC32 cells may represent a good model for studying the role of drug metabolism, in predicting therapeutic response and evaluating metabolic effects (Table 2).

Further, the effects of RIF and VD3, agonists of PXR and VDR, respectively, and known as inducers of CYP3A4 expression and activity [13] were investigated (Figure 5B–D). VD3 treatment resulted in the highest induction of CYP3A4. Compared to Caco-2, VD3 induction of CYP3A4 was higher in eight HROC lines (1.2- to 17-fold change, Figure 5B and Table 1). Among those, HROC183 T0 M2 showed the highest induction potential (17-fold at 72 h). Contrarily, induction of CYP3A4 via VD3 was lower in HROC217 T1 M2 cells than in Caco-2; however, an early onset of response was observed, which was stable over time (10-fold change at 48 h and 72 h). Low or no induction was observed in HROC43 and HROC60. RIF treatment increased CYP3A4 activity levels in two HROC lines above the level measured for RIF treated Caco-2 cells (1.4 to 5.1-fold change, Figure 5D). Greatest induction potential was observed in HROC217 T1 M2 (5.1-fold change at 48 h, 3.8-fold change at 72 h, respectively) and HROC183 T0 M2 (2.2-fold change at 24 h, 2-fold change at 72 h). An unexpected observation was a decreasing CYP3A4 activity in five HROC lines under RIF treatment. Up to 70% reduction in CYP3A4 activity was detected in HROC239 T0 M1 (at 24 h), HROC32 (at 72 h), and HROC43 (at 24 h and 72 h), respectively.

When summarizing the CYP3A4 induction analysis, HROC217 T1 M2 and HROC183 T0 M2 demonstrated higher RIF and VD3 CYP3A4 activity induction than Caco-2 cells. To our knowledge, a low-passaged human carcinoma cell line grown as a long lasting 2D monolayer and demonstrating PXR- and VDR-mediated CYP3A4 induction at levels comparable or superior to Caco-2 has not been described yet. Our data show that Caco-2 cells possess sufficiently strong PXR- and glucocorticoid receptor-mediated but only weak VDR-mediated CYP3A4 inducibility. Importantly, the expression levels are much higher in HROC183 T0 M2 and HROC217 T1 M2 compared to Caco-2. Whereas, glucocorticoid receptor-mediated CYP3A4 induction remains comparatively low in both lines. So far, human induced pluripotent stem cells-derived enterocyte-like cells have been found to be an appropriate alternative to Caco-2 cells and common 2D cell lines [20]. However, the initiation of the differentiation process is more time consuming and costly compared to 2D cultures. Other colonic carcinoma cell lines such as LS-180 cells were shown to induce CYP3A4 mRNA expression by VD3 (5-fold change) and RIF (3-fold change). Still, low TEER levels and higher permeability across cell layers of these cells hinder their broader application in drug discovery [19]. Therefore, HROC183 T0 M2 and HROC217 T1 M2 may be ideal screening tools to predict drug bioavailability of VDR- and PXR-ligands while maintaining barrier integrity (Table 2).

It should be noted that this study has several limitations. First, we did not investigate cellular differentiation including expression of enterocytic markers and tight junction proteins. However, for the latter, it has been shown, that TEER levels correlate with the strength of tight junctions between adjacent cells [29]. Second, the activity of various transporters, enzymes, and nuclear receptors other than CYP3A4 was not analyzed. More detailed investigations are needed to further validate the functional potential of HROC lines for ADME purposes.

One of the negative aspects of our results is the observation of decreasing CYP3A4 activity levels in five HROC lines after RIF treatment. It is known that a downregulation of drug metabolizing enzymes and changes in the pharmacokinetics of drugs may occur during infection, inflammation, and cancer [32]. Another limitation is the fact, that all HROC lines are adenocarcinomas and adapted to 2D growth. Therefore, many cellular properties will differ from those of small intestinal cells. However, these limitations also apply to Caco-2 cells, the gold standard 2D intestinal model. In addition to the data presented, we would like to stress the fact that this HROC cell line panel possesses some further advantages compared to Caco-2. They are available in low, even ultra-low passages, a feature which is known to minimize genetic alterations [33], and thus prevents high variation of TEER values with increasing passage numbers [33,34]. Moreover, depending on specific requirements, a HROC model with desired cancer driver mutations (Appendix A) might be especially useful for targeted approaches in drug discovery. The results obtained with these cells are highly reproducible. Handling of cells is cost-effective, and they are applicable for high-throughput screening. Thus, these cells may provide excellent alternatives to 3D cells.

In conclusion, and despite some limitations, the detailed analyses presented here imply that HROC183 T0 M2 and HROC217 T1 M2 might be excellent models for assessing intestinal first-pass metabolism of novel drug candidates and therefore provide reliable data for active receptor-mediated transport, especially via potential VDR ligands. Since VDR ligands have shown therapeutic potential in inflammation, osteoporosis, autoimmune diseases, and cancers [35], more appropriate models are needed for evaluation of novel VDR ligands.

## 3. Materials and Methods

### 3.1. Chemicals and Reagents

Rifampin (RIF) was purchased from Carl Roth (Karlsruhe, Germany) and vitamin D3 (VD3) from Hycultec (Beutelsbach, Germany). Fluorescein isothiocyanate–dextran 4000 (FD4) was purchased from Sigma Aldrich (Taufkirchen, Germany) and dissolved in PBS. Stock solution of RIF (121.5 mM) was prepared using DMSO as solvent and of VD3 (1 mM) using ethanol. Final DMSO concentrations were kept at a maximum of 0.1%.

### 3.2. Patient-Derived CRC Cells

Caco-2 cells were purchased from CLS (Eppelheim, Germany). The patient-derived HROC (Hansestadt Rostock, colorectal cancer) cell lines were taken from our large collection [21].

### 3.3. Cell Culture

Cells were cultured in DMEM/F12 (1:1) supplemented with 10% FCS and 2 mM L-glutamine (all cell culture reagents were from PAN-Biotech, Aidenbach, Germany, unless stated otherwise) and grown in a humidified 37 °C incubator with 5% CO_2_. Medium was changed every 2–4 days. Cells were harvested by washing with PBS and detaching with trypsin when reaching about 70% confluence. To minimize alterations in morphology, proliferation, and response to stimuli, cells were maximally used up to passage number 45 and all experiments were performed within a range of 20 passages maximum.

### 3.4. Monolayer Assessment

Cells (5 × 10^5^ cells per well) were seeded on 6-well plates (Sarstedt, Nümbrecht, Germany). To assess barrier-forming potential and long-term cultivation ability, cells were cultured for 21 days. Light microscopic images were taken when reaching 100% confluence. Cell lines which formed domes, developed gaps, or detached strongly after having reached confluence were excluded from further analyses.

### 3.5. Cell Proliferation and Migration

Cell proliferation data have either previously been published [36,37,38] or were assessed as follows: cells were seeded in 24-well plates (1 × 10^4^ to 4 × 10^4^ cells per well) and after a 48 h-attachment period, crystal violet staining was performed daily for four consecutive days. Absorbance was measured at 590 nm using the Tecan Infinite 200 Pro plate reader (Tecan, Männedorf, Switzerland). To assess cell migration, 5 × 10^5^ cells were seeded per well of a 6-well plate. When cells reached confluence, the medium was changed to 0% FCS and a scratch was administered. Photographs were taken daily after wounding for four consecutive days. Gap diameters were measured with ZEN core v.2.7 software (Zeiss, Oberkochen, Germany). Three independent measurements each were performed.

### 3.6. TEER Measurement

Cells were seeded in 24-well Transwell inserts (4 × 10^4^ cells per well) (ThinCerts, Greiner Bio-One, pore size = 8 µm) with 750 µL and 150 µL culture medium in the basolateral and apical compartment respectively. TEER values were measured at room temperature from day 1 (d1) to d21 post-seeding using the EVOM3 device (World Precision Instruments Europe, Friedberg, Germany). Mean values were calculated from three independent measurements. Blank inserts were used as controls. For final unit area resistance, raw data were multiplied by the effective area of the membrane (0.336 cm^2^).

### 3.7. Fluorescein Isothiocyanate–Conjugated Dextran 4000 (FD-4) Permeability Tests

Permeability tests were done at d3, d5, d7, d9, d14, and d21 post-seeding. Cell monolayers cultured on transwell inserts were washed with PBS; 500 µL phenol red-free DMEM and 150 µL FD-4 (0.4 mg/mL dissolved in PBS) were added to the basolateral and apical compartment, respectively. After 20 min incubation at room temperature, 100 µL from the basolateral compartment were transferred to white opaque 96-well microplates (Greiner Bio-one). Fluorescence intensity was measured at an excitation of 485 nm and emission of 535 nm. Blank values were subtracted from raw data and FD-4 concentration was calculated in percent of control wells without cells set to 100%. Data presented are from three independent experiments.

### 3.8. Luminometric Analysis of CYP3A4 Activity

CYP3A4 induction studies were performed by using the Promega CYP3A4/Luciferin-IPA P450-Glo™ Assay (Promega Cat No. V9001) with Luciferin-IPA, a highly sensitive and selective system for CYP3A4 activity detection. The non-lytic assay was done according to the manufacturer’s protocol. In addition, 20 µM RIF and 100 nM VD3 were used to evaluate CYP3A4 induction. Medium containing 0.1% DMSO (*v*/*v*) was used as vehicle control. Cells were seeded as duplicate in 96-well plates at a density of 10,000 cells/well. After reaching confluence, medium was replaced with medium containing the test compounds. This medium was refreshed daily. After incubation, medium was discarded and cells were washed with PBS before measurement. CYP3A4 activity was compared between control, 24 h, 48 h, and 72 h of drug treatment. Emitted luminescence was measured using the Tecan Infinite 200 Microplate Reader with integration time set to 1 s. Background luminescence was subtracted from all values and induction fold-changes were calculated by dividing the obtained values by those of the control. Data presented are from three independent experiments.

### 3.9. Data Analysis

The paired two tailed Student’s *t* test was used to perform statistical analysis between treated and untreated groups. GraphPad Prism software 9.0.1 was used for calculation.

## Figures and Tables

**Figure 1 ijms-23-09861-f001:**
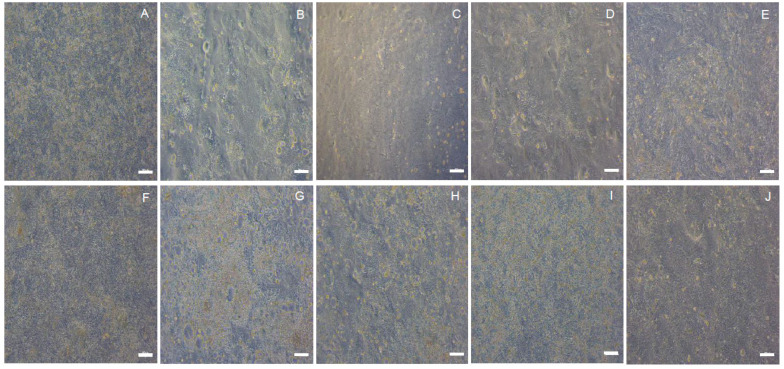
Light microscopy of HROC cell monolayers at 100% confluence. Scale bar = 100 µm, original magnification ×10: (**A**) HROC32, (**B**) HROC43, (**C**) HROC60, (**D**) HROC80 T1 M1, (**E**) HROC126, (**F**) HROC159 T2 M4, (**G**) HROC183 T0 M2, (**H**) HROC239 T0 M1, (**I**) HROC217 T1 M2 (**J**) HROC383.

**Figure 2 ijms-23-09861-f002:**
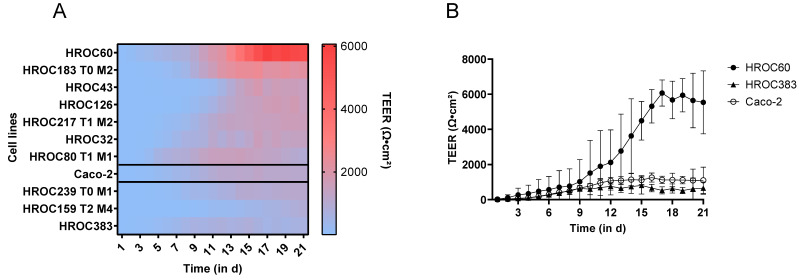
Measurement of intestinal epithelial barrier integrity by TEER. TEER measurement (in Ω·cm^2^) over a fixed surface area (0.336 cm^2^) detected for HROC- and Caco-2 cell layers grown on transwell inserts. (**A**) Heatmap of time-dependent TEER value alteration in HROC and Caco-2 cells and (**B**) HROC60 and HROC383 in comparison to Caco-2 cells, grown in Transwell inserts, measured from d1 to d21 post-seeding. Blank value was subtracted from raw data. Data are presented as mean (**A**) and mean ± SD (**B**), respectively, of three independent cultures.

**Figure 3 ijms-23-09861-f003:**
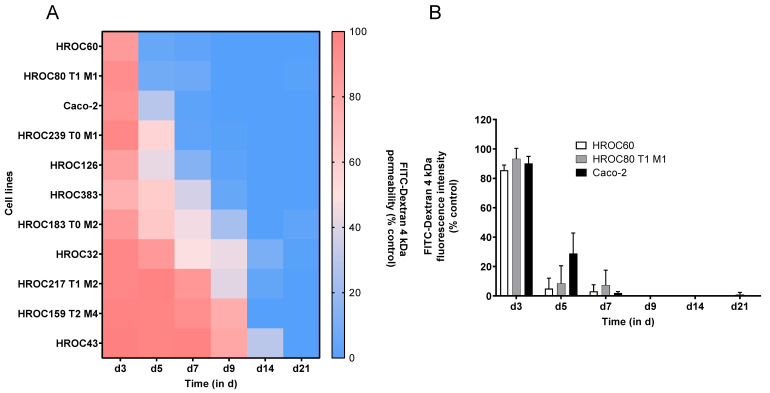
Measurement of intestinal epithelial barrier integrity by FD4 flux assay. Quantification of FITC-labeled dextran passing from apical to basolateral of HROC- and Caco-2 cell layers grown on transwell inserts. (**A**) Heatmap of time-dependent FD-4 permeability and (**B**) selected HROC cell lines in comparison to Caco-2 cells, measured on d3, d5, d7, d9, d14, and d21 post-seeding. The % FD-4 permeability was calculated as the amount of FD-4 in the basolateral compartment after 20 min incubation time divided by the total amount of FD-4 collected in empty cell free control multiplied by 100. Data represent mean of three independent cultures (**A**) and mean ± SD (**B**).

**Figure 4 ijms-23-09861-f004:**
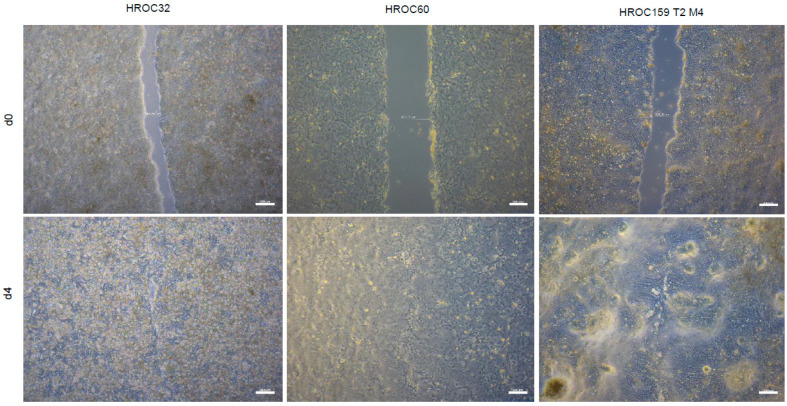
Scratch wound-healing assay of HROC cell lines. Representative images of selected HROC cell lines at d0 and d4 post-injury. Confluent cell monolayers were grown in 6-well plates maintained in media without FCS. Scratch was administered on d0, distances between the edges of the gap were recorded 96 h after wounding. Speed of wound closing (µm/h) was calculated by subtracting the initial distance from the distance of the last time point divided by the number of hours. Scale bar = 200 µm, original magnification × 4. Data represent mean ± SD.

**Figure 5 ijms-23-09861-f005:**
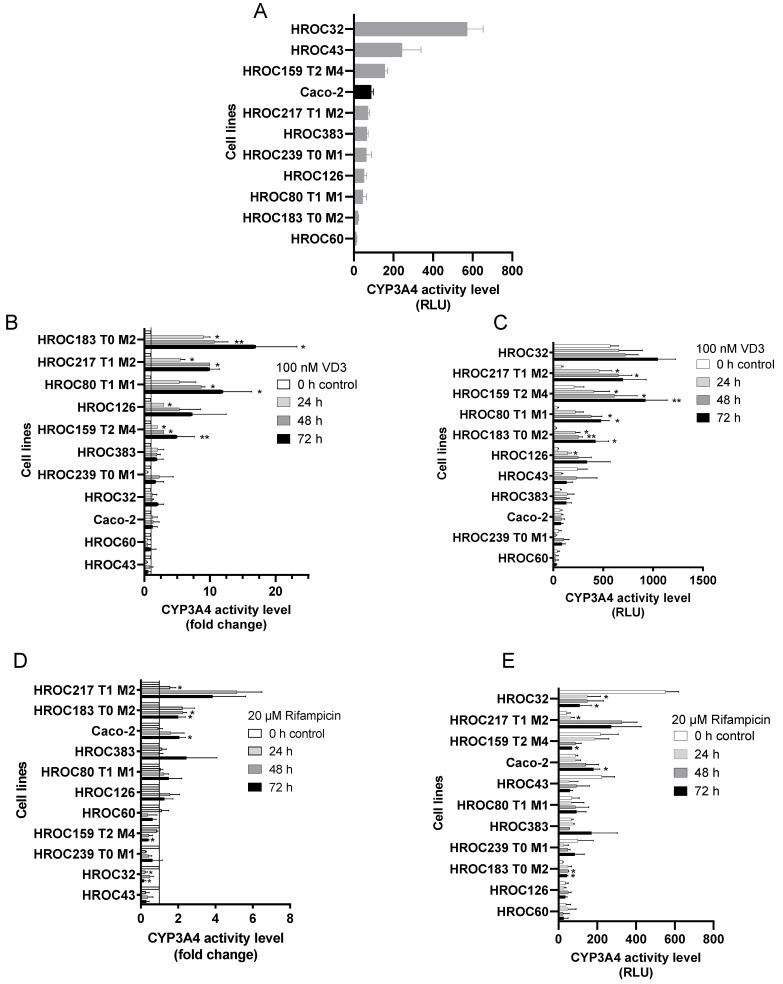
Basal CYP3A4 activity level and CYP3A4 induction potential in HROC and Caco-2 cells. Cells were treated with (**A**) 0.1% DMSO as vehicle control, (**B**,**C**) 100 nM VD3, and (**D**,**E**) 20 µM RIF. Fold-induction (**B**,**D**) was calculated from DMSO-treated cells, untreated control group was set at 1.0. For RLUs (**C**,**E**), blank value was subtracted from all values. Data are represented as the mean ± SD of three independent measurements. * *p* < 0.05, ** *p* < 0.01.

**Table 1 ijms-23-09861-t001:** Summarized table of results. Cell morphology, barrier forming capacity, and CYP3A4 induction potential in the HROC cell lines of the final panel in comparison to Caco-2.

	Barrier Forming Potency			CYP3A4 Inducibility	
Cell Line	TEER	FD-4 Permeability	Basal CYP3A4 Activity	PXR-Mediated	VDR-Mediated
HROC32	+	−	+++	−	++
HROC43	+	−	+++	−	−
HROC60	+++	− −	−	−	+
HROC80 T1 M1	+	− −	−	++	+++
HROC126	+	−	+	+	+++
HROC159 T2 M4	+	−	++	−	+++
HROC183 T0 M2	++	−	−	++	+++
HROC217 T1 M2	+	−	+	+++	+++
HROC239 T0 M1	+	−	+	−	++
HROC383	+	−	+	++	++
Caco-2	+	−	+	++	+

Maximum TEER levels are scored as “+” for 812–1259 Ω·cm^2^, “++” for 1547–2546 Ω·cm^2^, “+++” for ≥6000 Ω·cm^2^. For FD-4 permeability, “−” indicates drop of permeability comparable to Caco-2, “− −” indicates rapid decrease. Basal CYP3A4 activity levels compared to Caco-2 (0.6 to 1.4) are indicated by “+”, “++” indicate 2-fold higher (1.5 to 1.8) and “+++” 2.8- to 6.5-fold higher levels, whereas (−) indicates lower levels (up to 0.2). For CYP3A4 induction, “+” indicate fold changes from 0.7 to 1.3, “++” from 1.4 to 2.0 and “+++” ≥3.8, whereas (−) indicates decreasing levels (up to 0.3) after 72 h induction.

**Table 2 ijms-23-09861-t002:** Summarized table of potential applications for the intestinal model candidate HROC cell lines. Highlighted features and potential applications for the ten HROC cell lines of the final panel.

Cell Line	Feature	Application
HROC32	Very high basal CYP3A4 activity, VDR-mediated CYP3A4 inducibility	Drug absorption, development of alternative drug regimens, best model for evaluation of metabolic effects, VDR-mediated first-pass metabolism
HROC43	High basal CYP3A4 activity	Drug absorption, development of alternative drug regimens
HROC60	Extremely high barrier integrity, rapid barrier development, long-lasting monolayer	Drug absorption, best model for long-term experiments like ‘gut-on-a-chip’
HROC80 T1 M1	Rapid barrier development, VDR-mediated CYP3A4 inducibility	Drug absorption, VDR-mediated first-pass metabolism
HROC126	Rectal cancer cell line, VDR-mediated CYP3A4 inducibility	Drug absorption, Best model for rectal drug administration, VDR-mediated first-pass metabolism
HROC159 T2 M4	VDR-mediated CYP3A4 inducibility	Drug absorption, VDR-mediated first-pass metabolism
HROC183 T0 M2	High barrier integrity, high VDR-mediated CYP3A4 inducibility	Drug absorption, together with HROC217 T1 M2 best model for PXR-/VDR-mediated first-pass metabolism
HROC217 T1 M2	High VDR-mediated CYP3A4 inducibility	Drug absorption, together with HROC183 T0 M2 best model for PXR-/VDR-mediated first-pass metabolism
HROC239 T0 M1	Rectal cancer cell line, VDR-mediated CYP3A4 inducibility	Drug absorption, rectal drug administration, VDR-mediated first-pass metabolism
HROC383	PXR-mediated CYP3A4 inducibility	Drug absorption, PXR-mediated first pass metabolism

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
