# Peer review of "Establishment and Characterization of Novel Human Intestinal In Vitro Models for Absorption and First-Pass Metabolism Studies"

_ijms, 2022, doi:10.3390/ijms23179861_

Round 1

Reviewer 1 Report

Thank you for giving me the opportunity to review the article entitled: “Establishment and characterization of novel human intestinal in vitro models for absorption and first-pass metabolism studies”.

The study aimed to establish a platform of colorectal cancer patient-derived cell lines for evaluation of human intestinal drug absorption and metabolism. Ten 2D cell lines were characterized out of collection with confluent outgrowth and long-lasting barrier forming potential as well as suitability for high throughput applications with special emphasize on expression and inducibility of CYP3A4.

HROC183 T0 M2 and HROC217 T1 M2 cell lines were identified as useful tools for in vitro drug absorption studies. Due to their high TEER values and inducibility by drug receptor ligands, they may be superior to Caco-2 cells to analyze oral drug absorption and intestinal drug-drug interactions.

The authors mentioned that the study has limitations and more detailed investigations are needed to further validate the functional potential of HROC lines for ADME purposes. However, the HROC cell line panel possesses some further advantages compared to Caco-2. Moreover, HROC183 T0 M2 and HROC217 T1 M1 might be excellent models for assessing intestinal first-pass metabolism of novel drug candidates.

Minor changes in the conclusion are recommended to reflect the limitations of the study.

Author Response

We want to thank for this very positive feedback, and modified the conclusion according to this reviewers’ suggestion by adding “In conclusion, and despite some limitations, the detailed analyses…”.

Reviewer 2 Report

This manuscript aimed to identify colorectal cancer patient-derived cell lines that might be used for in vitro drug absorption studies and two such cell lines have been identified that could be especially useful for targeted approaches in drug discovery. Overall, this manuscript has a lot of qualities, much effort has been put in performing experiments and preparing the manuscript, it is generally well designed study, and the obtained results are new and significant. Besides, the limitations of the study have been fairly presented, along with the novelty and significance of the study.

In my opinion, there are only few concerns in the manuscript that need to be considered and clarified, mostly technical.

In the column for FD4 permeability in Table 1, “+” indicates drop of permeability comparable to Caco-2 and “++” indicates rapid decrease. However, in my opinion, it would be easier for the readers to understand if FD4 flux was presented different, to be clear that high TEER values are in direct correlation with low FD4 flux values.

In the section ’Drug-induced CYP3A4 activity’ it should be emphasized that rifampicin and vitamin D3 were used as agonists of PXR and VDR, respectively (it is mentioned earlier in the text, however it would be beneficial to state it here too).

There are several concerns on the use of symbols in the text. The multiplication sign should be the symbol ’×’ instead of ’*’. Besides, the percent sign and the number that corresponds to are not separated by space (e.g. line 148: 95-100%). Eventually, the concentrations should be expressed in a different manner. 1 mM is equivalent to 1 mmol/L. Therefore, in the lines 75, 76, 128 etc - instead of mM/L should be just mM.

Lines 68, 69 - parentheses not necessary.

Author Response

This reviewer was also very positive but mentioned several minor concerns, which we completely addressed:

  • The Table 1 was adapted according to this reviewers’ suggestions.
  • We modified the section “Drug-induced CYP3A4 activity” as follows: “Further, the effects of RIF and VD3, agonists of PXR and VDR, respectively and known as inducers of CYP3A4 expression…”.
  • The use of the symbols in the text has also been adapted as suggested. Of note, we also modified a legend in Figure 2 and accordingly uploaded a novel version of that figure.
  • In the lines 68 and 69, the parentheses were removed.